# Pooled prevalence and associated factors of health facility delivery in East Africa: Mixed-effect logistic regression analysis

**Getayeneh Antehunegn Tesema** *, **Zemenu Tadesse Tessema**

Department of Epidemiology and Biostatistics, Institute of Public Health, College of Medicine and Health Sciences, University of Gondar, Gondar, Ethiopia

* getayenehantehunegn@gmail.com

**Data Availability Statement:** The underlying data is available online from www.measuredhs.com.

**Funding:** The author(s) received no specific funding for this work.

## Abstract

### Background

Many mothers still give birth outside a health facility in Sub-Saharan Africa particularly in East African countries. Though there are studies on the prevalence and associated factors of health facility delivery, as to our search of literature there is limited evidence on the pooled prevalence and associated factors of health facility delivery in East Africa. This study aims to examine the pooled prevalence and associated factors of health facility delivery in East Africa based on evidence from Demographic and Health Surveys.

### Methods

A secondary data analysis was conducted based on the most recent Demographic and Health Surveys (DHSs) conducted in the 12 East African countries. A total weighted sample of 141,483 reproductive-age women who gave birth within five years preceding the survey was included. All analyses presented in this paper were weighted for the sampling probabilities and non-response using sampling weight (V005), primary sampling unit (V023), and strata (V021). The analysis was done using STATA version 14 statistical software, and the pooled prevalence of health facility delivery with a 95% Confidence Interval (CI) was presented using a forest plot. For associated factors, the Generalized Linear Mixed Model (GLMM) was fitted to consider the hierarchical nature of the DHS data. The Intra-class Correlation Coefficient (ICC), Median Odds Ratio (MOR), and Likelihood Ratio (LR)-test were done to assess the presence of a significant clustering effect. Besides, deviance (-2LLR) was used for model comparison since the models were nested models. Variables with a p-value of less than 0.2 in the bivariable mixed-effect binary logistic regression analysis were considered for the multivariable analysis. In the multivariable mixed-effect analysis, the Adjusted Odds Ratio (AOR) with 95% Confidence Interval (CI) were reported to declare the strength and significance of the association between the independent variable and health facility delivery.

### Results

The proportion of health facility delivery in East Africa was 87.49% [95% CI: 87.34%, 87.64%], ranged from 29% in Ethiopia to 97% in Mozambique. In the Mixed-effect logistic

**Competing interests:** The authors have declared that no competing interests exist.

**Abbreviations:** ANC, Antenatal Care; AOR, Adjusted Odds Ratio; CI, Confidence Interval; DHS, Demographic Health Survey; GLMM, Generalized Linear Mixed Models; ICC, Intra-class Correlation Coefficient; LLR, log-likelihood Ratio; LR, Likelihood Ratio; MOR, Median Odds Ratio; SSA, Sub-Saharan Africa; WHO, World Health Organization.

regression model; country, urban residence [AOR = 2.08, 95% CI: 1.96, 2.17], primary women education [AOR = 1.61, 95% CI: 1.55, 1.67], secondary education and higher [AOR = 2.96, 95% CI: 2.79, 3.13], primary husband education [AOR = 1.19, 95% CI: 1.14, 1.24], secondary husband education [AOR = 1.38, 95% CI: 1.31, 1.45], being in union [AOR = 1.23, 95% CI: 1.18, 1.27], having occupation [AOR = 1.11, 95% CI: 1.07, 1.15], being rich [AOR = 1.36, 95% CI: 1.30, 1.41], and middle [AOR = 2.14, 95% CI: 2.04, 2.23], health care access problem [AOR = 0.76, 95% CI: 0.74, 0.79], having ANC visit [AOR = 1.54, 95% CI: 1.49, 1.59], parity [AOR = 0.56, 95% CI: 0.55, 0.61], multiple gestation [AOR = 1.83, 95% CI: 1.67, 2.01] and wanted pregnancy [AOR = 1.19, 95% CI: 1.13, 1.25] were significantly associated with health facility delivery.

## Conclusion

This study showed that the proportion of health facility delivery in East African countries is low. Thus, improved access and utilization of antenatal care can be an effective strategy to increase health facility deliveries. Moreover, encouraging women through education is recommended to increase health facility delivery service utilization.

## Background

Maternal and child mortality remains a major public health problem in low-and middle-income countries mainly in Sub-Saharan Africa (SSA) countries [1]. Globally, an estimated 358,000 maternal deaths occur annually, of which 99% occurred in low-and middle-income countries [2]. Even though the maternal mortality rate showed a substantial reduction in high-income countries [3], SSA continues to share the huge burden of global maternal mortality [4].

The World Health Organization (WHO) recommends health facility delivery as a key strategy to reduce maternal and infant mortality [5, 6]. According to the WHO, every pregnant woman should give birth at a health facility but only 48% of SSA births are delivered in a health facility [7, 8]. The lower proportion of health facility delivery is the reflection of poor affordability and accessibility of maternal health care services [9, 10].

Several studies found health facility delivery as a significant predictor responsible for the reduction of maternal and neonatal mortality [11, 12]. However, access to and use of maternal health care services in East African countries remains a major challenge [13, 14]. Previous revealed that maternal education [15], household wealth status [16], maternal occupation [17], husband education [18], distance to health facility [19], residence [20], parity [21], maternal age [20], marital status [22], Antenatal Care (ANC) visit during pregnancy [23, 24], type of gestation [24], and wanted pregnancy [25] were significantly associated factors with health facility delivery.

SSA is projected to have at least 80% of deliveries in health facilities particularly in East African countries, given international success in reducing maternal and neonatal mortality [26]. To improve health facility delivery in East African countries multisectoral collaboration is needed and international stakeholders might work on common factors responsible for the reduction of health facility delivery in different countries.

As far as our literature search is concerned, little is known about the pooled prevalence of health facility delivery and associated factors in East African countries. Therefore, this study aimed at investigating the pooled prevalence and associated factors of health facility delivery in

East African Countries based on the most recent Demographic and Health Surveys (DHSs). The findings of this study may aid in the development of evidence-based public health policies to reduce maternal and newborn mortality. Furthermore, since this was a pooled analysis, the study power was increased, allowing for a thorough examination of effect modification within the data.

## Methods

### Data source and sampling procedures

The DHS data of 12 East African countries (Burundi, Ethiopia, Comoros, Uganda, Rwanda, Tanzania, Mozambique, Madagascar, Zimbabwe, Kenya, Zambia, and Malawi) [27] were used for this study. The DHS is a nationally representative survey that contains data on health and health-related indicators like mortality, morbidity, family planning service utilization, fertility, maternal and child health. The variables were extracted based on literature and appended together to determine the pooled prevalence and associated factors of health facility delivery in East Africa. The DHS employed a two-stage stratified sampling technique to select the study participants. In the first stage, Enumeration Areas (EAs) were randomly selected while in the second stage households were selected. Each country's survey consists of different datasets including men, women, children, birth, and household datasets, and for this study, we used the women's datasets (IR file). A total weighted sample of 141,483 reproductive-age women who gave birth in the last five years preceding the survey was included in this study (Table 1).

### Study variables and measurements

The outcome variable for this study was the place of delivery. For mothers who had more than one child in the last five years preceding the survey, the most recent birth was selected. Place of delivery was categorized into home delivery (when the birth took place at home) or health facility delivery (when the birth took place at the hospital, health center, or health post). The response variable for the $i^{th}$ mother was represented by a random variable Yi with two possible values coded as 1 and 0. So, the response variable of the $i^{th}$ mother Yi was measured as a dichotomous variable with possible values Yi = 1, if ith mother gave birth at the health facility, and Yi = 0 if a mother gave birth at home delivery.

**Table 1. The number of study participants in this study.**

| Country | Number of reproductive age women who gave birth within 5 years preceding the survey | | Study year |
|---|---|---|---|
| | **Home delivery** | **Health facility delivery** | |
| Burundi | 1480 | 12131 | 2016/17 |
| Comoros | 604 | 2276 | 2012 |
| Ethiopia | 7809 | 3213 | 2016 |
| Kenya | 6436 | 13037 | 2014 |
| Madagascar | 7013 | 5394 | 2008/09 |
| Malawi | 791 | 16604 | 2015/16 |
| Mozambique | 285 | 11192 | 2011 |
| Rwanda | 537 | 7462 | 2014/15 |
| Tanzania | 3042 | 7010 | 2015/16 |
| Uganda | 2944 | 12326 | 2016 |
| Zambia | 3778 | 9564 | 2018 |
| Zimbabwe | 833 | 5585 | 2015 |

The independent variable retrieved from DHS were country, residence, maternal age, occupational status, women's educational status, husband's educational status, wealth status, distance to health care access, ANC visit during the index pregnancy, parity, marital status, preceding birth interval, number of gestation, and wanted pregnancy (Table 2). As the DHSs of the 12 East African countries were not conducted at the same time, we considered the year of the survey as an independent variable by considering 2008 as a reference. The Year of the survey was categorized as 2008 (Madagascar), 2011 (Mozambique), 2012 (Malawi), 2014 (Rwanda and Kenya), 2015 (Malawi, Tanzania, and Zimbabwe), 2016 (Burundi, Uganda, and Ethiopia), and 2018 (Zambia). However, the bi-variable analysis has a p-value of >0.2 and was not eligible for the multivariable analysis.

## Data management and analysis

We pooled the data from the 12 East African countries together after extracting the variables based on literature. Before any statistical analysis, the data were weighted using sampling weight (V005), primary sampling unit (V023), and strata (V021) to draw a valid conclusion. Data management and analysis were done using STATA version 14 statistical software. The pooled proportion of health facility delivery with the 95% Confidence Interval (CI) was reported using a forest plot. The hierarchical nature of DHS data could violate the independence of observations and equal variance assumption of the traditional logistic regression model. In such cases, advanced statistical models should be fitted to get a reliable estimate. Therefore, a mixed effect logistic regression model (fixed and random effect) was fitted using a cluster variable (V001) as a random variable. The presence of clustering effect was tested using the Intra-class Correlation Coefficient (ICC), Likelihood Ratio (LR) test, and Median Odds Ratio (MOR), and model comparison was made using deviance (-2LLR).

**Table 2. The list of independent variables and their definitions and measurements.**

| Variable name | Definition (measurement) |
|---|---|
| Country | Was coded as 0 "Burundi", 1 "Comoros", 2 "Ethiopia", 3 "Kenya", 4 "Madagascar", 5 "Malawi", 6 "Mozambique", 7 "Rwanda", 8 "Tanzania", 9 "Uganda", 10 "Zambia" and 11 "Zimbabwe" |
| Residence | Recoded as 0 for rural and 1 for urban |
| Age of respondent | Categorized as 0 for 15–24 years, 1 for 25–34 years and 2 for 35–49 years |
| Occupational status | Women occupation was No "if women were housewife and didn't working", and Yes "If a woman were working, she might be self-employed or government employed" |
| Maternal education status | Categorized as; didn't have formal education, attained primary level of education, and "secondary education and above" |
| Husband education | Categorized as; didn't have formal education, attained primary level of education, and "secondary education and above" |
| Wealth status | Categorized as; poor "if woman was in poorer and poorest household", middle and rich "if woman was in richer and richest household" |
| Preceding birth interval | Was categorized as; less than 24 months and ≥ 24 months |
| ANC visit during pregnancy | Categorized as No "if woman didn't have ANC visit during pregnancy" and Yes "If women had at least one ANC visit during pregnancy" |
| Distance to reach health facility | Categorized as;" A big problem" and "not a big problem" |
| Parity | Number of ever born children after 28 months of gestations categorized as; 1 birth, 2–4 birth, and ≥ 5 births |
| Number of gestations | Categorized as; single and multiple birth |
| Wanted pregnancy | Categorized as Yes "if wanted" and No "if mistimed or unwanted" |
| Marital status | Categorized as; not in union "if never married, divorced, widowed or separated" and in union "if married" |

The ICC quantifies the degree of heterogeneity of health facility delivery between clusters (the proportion of the total observed individual variation in health facility delivery that is attributable to between cluster variations) [28].

$$ICC = \sigma^2/(\sigma^2 + \pi^2/3).$$

MOR quantifies the variation or heterogeneity in health facility delivery between clusters and is defined as the median value of the odds ratio between the cluster at a high likelihood of health facility delivery and cluster at lower risk when randomly picking out two clusters (EAs) [29].

$$MOR = \exp{(\sqrt{2 * \partial 2 * 0.6745})} \sim MOR = \exp{(0.95*\partial)}.$$

$\partial^2$ indicates that cluster variance.

Variables with a p-value <0.2 in the bi-variable analysis were considered in the multivariable mixed-effect logistic regression analysis. In the multivariable mixed-effect logistic regression model the Adjusted Odds Ratios (AOR) with a 95% Confidence Interval (CI) were reported to declare the statistical significance and strength of association between factors and health facility delivery.

## Ethics consideration

Permission to get access to the data was obtained from the measure DHS program online request from http://www.dhsprogram.com.website and the data used were publicly available with no personal identifier.

## Results

### Socio-demographic and economic characteristics of the respondent

A total of 141,483 live births were included. Of these, 19,563 (13.8%) births were from Kenya, and 110,471 (78.1%) were in rural areas. Besides, 67,704 (47.9%) births were born to mothers aged 25–34 years. More than half (53.7%) of mothers and 41.9% of their husbands attained primary level of education (Table 3).

### Maternal obstetric and health services elated characteristics of the respondent

From a total of 141,483 births, 22,141 (15.7%) of the mothers were primipara, and 4,504 (3.2%) gave multiple births. About 93,360 (66.0%) of the mothers had ANC follow-up during pregnancy and 121,189 (85.7%) were wanted births. Regarding health care access, the majority (57.1%) of the mothers reported distance to reach a health facility as a big problem (Table 4).

### The pooled prevalence of institutional delivery in East African countries

The pooled proportion of health facility delivery in East African countries was 87.49% [95% CI: 87.34, 87.64], with the highest proportion in Mozambique (97%) and the lowest proportion in Ethiopia (29%) (Fig 1).

### Factors associated with health facility delivery

**Model comparison.** The mixed-effect logistic regression model was the best-fitted model since it had a smaller deviance value (Table 5). Furthermore, the ICC value was 0.22 [95% CI: 0.21, 0.24] and MOR was 2. 52, it indicates if we randomly choose two women from different

**Table 3. Socio-demographic and economic characteristics of women who gave birth in the last five years in East African countries.**

| Characteristics | Weighted frequency | Percentage (%) |
|---|---|---|
| **Country** | | |
| Burundi | 13,611 | 9.6 |
| Comoros | 2,880 | 2.0 |
| Ethiopia | 11,022 | 7.8 |
| Kenya | 19,563 | 13.8 |
| Madagascar | 12,407 | 8.8 |
| Malawi | 17,395 | 12.3 |
| Mozambique | 11,478 | 8.1 |
| Rwanda | 8,002 | 5.7 |
| Tanzania | 10,052 | 7.1 |
| Uganda | 15,270 | 10.8 |
| Zambia | 13,383 | 9.5 |
| Zimbabwe | 6,418 | 4.5 |
| **Residence** | | |
| Urban | 31,012 | 21.9 |
| Rural | 110,471 | 78.1 |
| **Age in years** | | |
| 15–24 | 42,167 | 29.8 |
| 25–34 | 67,704 | 47.9 |
| $\geq 35$ | 31,612 | 22.3 |
| **Women education status** | | |
| No | 33,619 | 23.8 |
| Primary | 75,945 | 53.7 |
| Secondary and above | 31,907 | 22.5 |
| **Husband education** | | |
| No | 25,268 | 17.9 |
| Primary | 59,332 | 41.9 |
| Secondary and above | 56,882 | 40.2 |
| **Women occupational status** | | |
| No | 47,153 | 33.3 |
| Yes | 94,330 | 66.7 |
| **Marital status** | | |
| Not in union | 41,222 | 29.1 |
| In union | 100,261 | 70.9 |
| **Wealth status** | | |
| Poor | 64,368 | 45.5 |
| Middle | 27,586 | 19.5 |
| Rich | 49,529 | 35.0 |

clusters, a woman from a cluster with higher health facility delivery were 2. 52 times more likely to deliver at a health facility than women from a cluster with a lower proportion of health facility. Besides, the likelihood ratio test was (LR test vs. Logistic model: $X^2 (01) = 6623.18$, p<0.01) which informed that the mixed-effect logistic regression model is the better model over the basic model (Table 5).

In the multivariable mixed-effect logistic regression model; country, residence, maternal and husband educational status, marital status, wealth status, maternal occupation, distance to

**Table 4. Maternal obstetric and health service-related characteristics of the respondent.**

| Characteristics | Weighted frequency | Percentage (%) |
|---|---|---|
| **Parity** | | |
| 1 | 22,141 | 15.7 |
| 2–4 | 72,858 | 51.5 |
| ≥ 5 | 46,484 | 32.8 |
| **Number of gestations** | | |
| Single | 136,979 | 96.8 |
| Multiple | 4,504 | 3.2 |
| **Preceding birth interval** | | |
| < 24 months | 19,380 | 17.9 |
| ≥ 24 months | 88,934 | 82.1 |
| **Health care access problem** | | |
| No a big problem | 60,714 | 42.9 |
| A big problem | 80,769 | 57.1 |
| **Wanted pregnancy** | | |
| No | 20,294 | 14.3 |
| Yes | 121,189 | 85.7 |
| **ANC visit during pregnancy** | | |
| No | 48,123 | 34.0 |
| Yes | 93,360 | 66.0 |

ANC: Antenatal Care.

the health facility, ANC visit during pregnancy, and wanted pregnancy was significantly associated with health facility delivery.

Mothers in Burundi, Kenya, Comoros, Malawi, Mozambique, Rwanda, Tanzania, Uganda, Zambia and Zimbabwe were 18.04 [AOR = 18.04, 95% CI: 16.65, 19.53], 1.97 [AOR = 1.97, 95% CI: 1.83, 2.12], 6.14 [AOR = 6.14, 95% CI: 5.48, 6.88], 36.99 [AOR = 36.99, 95% CI: 33.69, 40.62], 82.81 [AOR = 82.81, 95% CI: 72.66, 94.36], 22.89 [AOR = 22.89, 95% CI: 20.49, 25.58], 3.63 [AOR = 3.41, 95% CI: 3.18, 3.66], 6.67 [AOR = 6.67, 95% CI: 6.21, 7.16], 3.41 [AOR = 3.41, 95%: 3.18, 3.66] and 6.05 [AOR = 6.05, 95% CI: 5.46, 6.71] times higher odds of having health facility delivery compared to mothers in Ethiopia, respectively. Mothers who lived in urban area were 2.08 times [AOR = 2.08, 95% CI: 1.96, 2.17] higher odds of having health facility delivery than rural mothers.

Mothers who attained primary education, and secondary or above were 1.61 times [AOR = 1.61, 95% CI: 1.55, 1.67] and 2.96 times [AOR = 2.96, 95% CI: 2.79, 3.13] higher odds of having health facility delivery compared to mothers who did not have formal education, respectively. Mothers whose husband had a primary level of education, and secondary education or above were 1.19 times [AOR = 1.19, 95% CI: 1.14, 1.24] and 1.38 times [AOR = 1.38, 95% CI: 1.31, 1.45] higher odds of giving birth at a health facility than mother whose husband did not have formal education, respectively.

The odds of health facility delivery utilization by women who were in union were 1.23 times [AOR = 1.23, 95% CI: 1.18, 1.27] higher than women who were not in union, and mothers who had occupation were 1.11 times [AOR = 1.11, 95% CI: 1.07, 1.15] higher odds of having health facility than women who did not have an occupation. Women from a household with middle and rich wealth status were 1.36 [AOR = 1.36, 95% CI: 1.30, 1.41], and 2.14 [AOR = 2.14, 95% CI: 2.04, 2.23] times higher odds of health facility delivery than women from a poor household, respectively. The odds of health facility delivery among mothers who

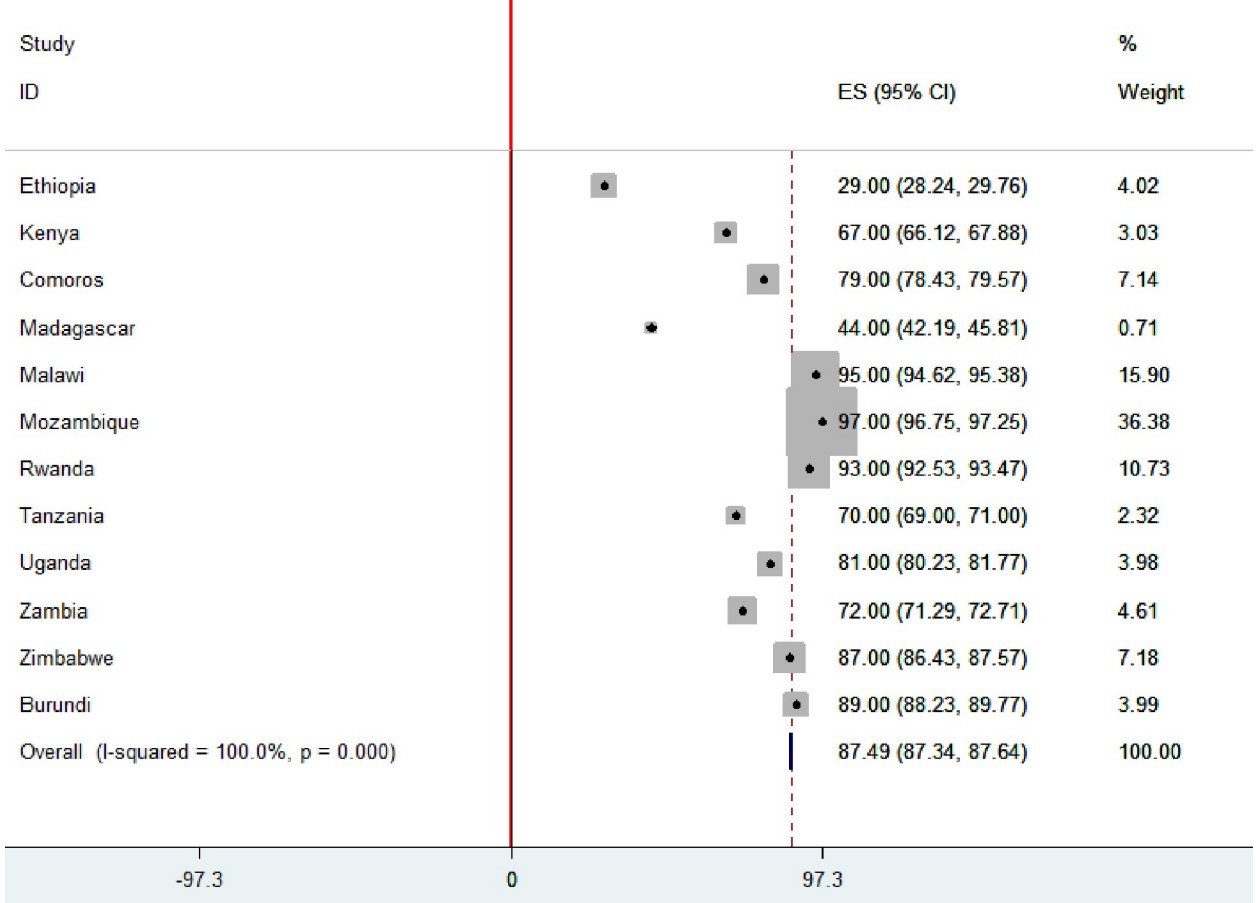

**Fig 1. The proportion of health facility delivery in East African countries.**

had a big health care access problem were decreased by 24% [AOR = 0.76, 95% CI: 0.74, 0.79] compared to women where health care access was not a big problem.

Mothers who had ANC follow-up during pregnancy had 1.54 [AOR = 1.54, 95% CI: 1.49, 1.59] times higher odds of health facility delivery than women who did not have ANC visit. The odds of health facility delivery among mothers who had two to four births, and five and above were decreased by 44% [AOR = 0.56, 95% CI: 0.55, 0.61] and 62% [AOR = 0.38, 95% CI:

**Table 5. Model comparison and random effect results.**

| Parameter | Standard logistic regression | Mixed-effect logistic regression analysis (GLMM) |
|---|---|---|
| LLR | -56792 | -55922 |
| Deviance | 113584 | 110844 |
| ICC | 0.22 [0.21, 0.24] | |
| LR-test | LR test vs. logistic model: chibar2(01) = 6623.18 Prob > = chibar2 <0.001 | |
| MOR | 2.52 [95% CI:2.41, 2.63] | |
| Cluster variance | 0.95 [95% CI: 0.86, 1.04] | |

*LLR; log-likelihood ratio, ICC; Intra-class Correlation Coefficient, MOR; Median Odds Ratio, LR-test; Likelihood Ratio test.

0.35, 0.40] compared to women who were primipara, respectively. Mothers with multiple gestations were 1.83 [AOR = 1.83, 95% CI: 1.67, 2.01] times higher odds of delivering at a health facility compared to mothers with single gestation, and mothers whose pregnancy were wanted to have 1.19 [AOR = 1.19, 95% CI: 1.13, 1.25] times increased odds of health facility delivery (Table 6).

## Discussion

In East Africa, the proportion of health facility delivery was 87.49%. This was higher than a study reported in SSA [8]. The possible explanation could be due to the difference in the study period and the number of countries included in the study. It was significantly varied across counties, ranging from 29% in Ethiopia to 97% in Mozambique. This may be because of the lack of sufficient medical care and human resources in the Ethiopian health system to satisfy more than 100 million Ethiopians [30].

In the multivariable mixed effect binary logistic regression analysis; country, residence, maternal education status, husband education status, marital status, household wealth status, ANC visit during pregnancy, wanted pregnancy, health care access problem, parity, number of gestations, and occupational status were significantly associated with health facility delivery. In this study, the mother's place of residence was significantly associated with health facility delivery. Urban mothers had higher odds of having health facility delivery than rural mothers. This is consistent with studies reported in SSA [31] and Africa [32]. This might be due to the residential disparity in availability and accessibilities of maternal health care services [33, 34]. Also, in urban areas maternal education [35], access to maternal health services [36], and access to information is relatively good than rural mothers [17, 34]. Furthermore, evidence suggests that the majority of women in rural areas preferred to give birth at home with traditional birth attendants for the sake of privacy and social acceptance than urban mothers [6].

Women who attained primary education or higher were more likely to give birth at a health facility than women with no formal education. It was in line with studies reported in Sub-Saharan Africa [31], China [37], and the WHO Global survey [38]. This may be because educated mothers would have a better understanding of the risks of childbirth, the provision of maternal health care, and the value of health facility delivery for newborns and their health [39]. In addition, maternal education plays a significant role in enhancing the mother's health care decision-making autonomy [40]. Besides, husband education was a significant predictor of health facility delivery, women whose husbands completed primary education or higher were more likely to deliver in the health facility than women whose husbands did not have formal education. It was supported by previous studies reported in Pakistan [18], Nigeria [41], Nepal [42], and Sub-Saharan Africa [43]. This might be since educated husbands can empower women in making health care decisions and get involved in making birth preparedness and complication preparedness plan would also increase the center's service use [44]. Besides, educated men would have better access to information about the importance of health facility delivery and complications of home delivery to the mother and their baby [45], and in fact, education leads to better health awareness, which may sensitize the mother to decide and utilize maternal health care services [46].

Being married had higher odds of health facility delivery than women who were not in a union. It was consistent study findings in Kenya [47] and Nigeria [41]. This might be due to married women had spousal support in making health care decisions towards maternal health service utilization as well as economic and social support [48]. Besides, when women pregnant without being in union are possibly less motivated to give birth at a health facility due to community stigmatization and marginalization [49].

**Table 6. Multivariable mixed-effect logistic regression analysis of determinants of health facility delivery in East African countries.**

| Variable | Place of delivery | | Crude Odds Ratio (COR) with 95% CI | Adjusted Odds Ratio (AOR) with 95% CI |
|---|---|---|---|---|
| | Home | Health facility | | |
| **Country** | | | | |
| Burundi | 1480 | 12131 | 16.82 [15.67, 18.05] | 18.04 [16.65, 19.53] |
| Comoros | 604 | 2276 | 5.56 [5.01, 6.16] | 1.97 [1.83, 2.12] |
| Ethiopia | 7809 | 3213 | 1 | 1 |
| Kenya | 6437 | 13037 | 3.38 [3.19, 3.59] | 6.14 [5.48, 6.88] |
| Madagascar | 7013 | 5394 | 1.48 [1.40, 1.57] | 1.01 [0.94, 1.08] |
| Malawi | 791 | 16604 | 50.31 [46.08, 54.93] | 36.99 [33.69, 40.62] |
| Mozambique | 286 | 11192 | 77.15 [68.05, 87.47] | 82.81 [72.66, 94.36] |
| Rwanda | 537 | 7462 | 29.47 [26.63, 32.62] | 22.89 [20.49, 25.58] |
| Tanzania | 3042 | 7010 | 4.75 [4.47, 5.05] | 3.63 [3.37, 3.90] |
| Uganda | 2945 | 12326 | 7.78 [7.34, 8.25] | 6.67 [6.21, 7.16] |
| Zambia | 3778 | 9564 | 5.31 [5.01, 5.62] | 3.41 [3.18, 3.66] |
| Zimbabwe | 833 | 5585 | 14.25 [13.02, 15.60] | 6.05 [5.46, 6.71] |
| **Residence** | | | | |
| Urban | 2997 | 22977 | 3.55 [3.41, 3.70] | 2.08 [1.96, 2.17] |
| Rural | 32557 | 77818 | 1 | 1 |
| **Maternal education status** | | | | |
| No | 13920 | 19679 | 1 | 1 |
| Primary | 18482 | 57388 | 2.54 [2.47, 2.62] | 1.61 [1.55, 1.67] |
| Secondary and above | 3147 | 28722 | 7.16 [6.84, 7.48] | 2.96 [2.79, 3.13] |
| **Husband education status** | | | | |
| No | 9797 | 15462 | 1 | 1 |
| Primary | 15623 | 43677 | 2.10 [2.04, 2.17] | 1.19 [1.14, 1.24] |
| Secondary and above | 10134 | 46657 | 3.25 [3.14, 3.37] | 1.38 [1.31, 1.45] |
| **Respondent age in years** | | | | |
| 15–24 | 8908 | 33,219 | 1 | 1 |
| 25–34 | 17026 | 50611 | 0.78 [0.76, 0.81] | 1.05 [0.97, 1.10] |
| ≥35 | 9620 | 21965 | 0.60 [0.58, 0.62] | 1.05 [0.99, 1.11] |
| **Marital status** | | | | |
| Not in union | 8256 | 32943 | 1 | 1 |
| In union | 27298 | 72852 | 0.63 [0.61, 0.65] | 1.23 [1.18, 1.27] |
| **Maternal occupation status** | | | | |
| No | 12774 | 34277 | 1 | 1 |
| Yes | 22780 | 71518 | 1.26 [1.23, 1.30] | 1.11 [1.07, 1.15] |
| **Wealth status** | | | | |
| Poor | 22262 | 42031 | 1 | 1 |
| Middle | 7075 | 20485 | 1.83 [1.77, 1.90] | 1.36 [1.30, 1.41] |
| Rich | 6217 | 43279 | 4.43 [4.28, 4.58] | 2.14 [2.04, 2.23] |
| **Distance to health care access** | | | | |
| Not a big problem | 19282 | 66007 | 1 | 1 |
| Big problem | 16272 | 39788 | 0.71 [0.69, 0.73] | 0.76 [0.74, 0.79] |
| **ANC visit during pregnancy** | | | | |
| No | 17064 | 30943 | 1 | 1 |
| Yes | 18490 | 74852 | 2.27 [2.22, 2.33] | 1.54 [1.49, 1.59] |
| **Parity** | | | | |
| 1 | 2748 | 19390 | 1 | 1 |

(*Continued*)

**Table 6.** (Continued)

| Variable | Place of delivery | | Crude Odds Ratio (COR) with 95% CI | Adjusted Odds Ratio (AOR) with 95% CI |
|---|---|---|---|---|
| | Home | Health facility | | |
| 2–4 | 16106 | 56683 | 0.50 [0.48, 0.52] | 0.56 [0.55, 0.61] |
| ≥ 5 | 16701 | 29721 | 0.25 [0.24, 0.27] | 0.38 [0.35, 0.40] |
| **Number of gestations** | | | | |
| Single | 34689 | 102167 | 1 | 1 |
| Multiple | 865 | 3628 | 1.47 [1.36, 1.59] | 1.83 [1.67, 2.01] |
| **Wanted pregnancy** | | | | |
| No | 6661 | 14031 | 1 | 1 |
| Yes | 29394 | 91763 | 1.36 [1.31, 1.41] | 1.19 [1.13, 1.25] |

Household wealth status and women's occupation were found to be significant predictors of health facility delivery. Women who were from the household with middle and rich wealth stratus were more likely to give birth at a health facility than women who were from poor households. It was consistent with the study findings in Nepal [50] and SSA [31]. In some African countries such as Ethiopia, Tanzania, and Kenya, even though maternal health services are offered free of charge by law or pro-poor fee exemption [51], indirect costs such as transportation costs and other opportunity costs for mothers and newborns prohibit mothers from using health facility delivery from poor families [52].

In this study, parity was an important predictor of health facility delivery. Primiparous women had higher odds of health facility delivery than multi-parous women. This finding was supported by previous studies [20, 21, 41], it could be because primigravida women feel that they are more prone to complications during delivery and seek maternity care services [53]. Besides, Multiparous women often choose to give birth at home for the sake of privacy and feel they will not be complicated as they are familiar with childbirth [54]. Besides, multiparous women are the least likely to seek maternity care services due to greater confidence and cumulative experience of delivery [54]. Having an ANC visit during pregnancy increases the likelihood of health facility delivery than women who did not have ANC visits during pregnancy. It was consistent with previous studies [18, 55], this is attributed to the assumption that mothers who have ANC visit during pregnancy may increase women's knowledge of birth preparedness and risks of pregnancy and childbirth, which may increase the possibility of getting delivery at health facilities [56, 57]. Furthermore, the use of ANC may signify the availability of a nearby health care service, which may also provide delivery care, and ANC providers should educate and advise women and their families on danger signs through a process to create individual birth plans that can prepare them for institutional delivery and make timely decisions in the event of an emergency to pursue health care [17]. Mothers whose pregnancy was wanted to have higher odds of health facility delivery than an unwanted pregnancy. When pregnancy is wanted, pregnant women may have ANC visits and regular medical check-ups, which could increase their knowledge of potential complications and safe delivery practices, ultimately encouraging them to pursue health facility delivery to get a healthy child [58].

The other most significant predictor of health facility delivery in this study was the type of gestation. Mothers who have multiple gestations had higher odds of health facility delivery than singletons. This was consistent with prior studies [24, 59], This may be because mothers with multiple gestations are at greater risk of complications related to pregnancy, such as obstructed labor, birth asphyxia, antepartum hemorrhage, preeclampsia, and postpartum hemorrhage [60], which may encourage women to give birth in a health facility. Besides, the health care access problem was associated with a lower likelihood of health facility delivery.

This could be due to the reason that the health care access problem is the main factor for home delivery, it highlights that there is a need to make maternal health care services available and accessible to the community.

Though enhancing health facility delivery is identified as the best strategy to achieve the Sustainable Development Goal (SDG) 3 of ending preventable maternal mortality and reducing Maternal Mortality Ratio (MMR) to fewer than 70 maternal death per 100,000 live births by 2030, health facility delivery was low in East African countries. Therefore, the stakeholders, governmental and non-governmental organizations should promote health facility delivery through enhancing mothers' ANC service utilization, and promoting women's education. Besides, special emphasis should be given to rural residents and poor households to improve health facility delivery. Even though the World Health Organization's recommendation for every pregnant woman to give birth at the health facility and by a skilled birth attendant, the rate remains low in East African countries. This may be because women need effective support to decrease home delivery practice. It is therefore important to improve women, family, and community awareness about institutional delivery, ANC follows up, and knowledge of danger signs of pregnancy.

## Strength and limitations

The strength of this study was that it was based on a weighted large, nationally representative data set and could have adequate statistical power to detect the true association of factors with health facility delivery. Besides, the study is done using an advanced model to take into account the clustering effect (mixed-effect logistic regression) to get reliable standard error and estimate. However, the study finding is interpreted in light of limitations. First, as with other cross-sectional studies, the temporal relationship can't be established. Second, the DHS didn't incorporate information about health care availability and accessibility like distance to the health facility, and the quality of maternal health services provided which might influence the use of health facility delivery of reproductive-age women. Also, since data was collected from self-report from respondents there may be a possibility of social desirability bias.

## Conclusions

This study found that health facility delivery in East Africa was far below to achieve a sustainable development goal. Country, urban residence, maternal education, husband education, multiple gestations, wanted pregnancy, ANC visit during pregnancy, middle and rich wealth status, having an occupation, and married marital status was positively associated with health facility delivery. Whereas, multiparty, and big health care access problem was negatively associated with health facility delivery. Thus, improved access and utilization of antenatal care can be an effective strategy to increase health facility deliveries. Moreover, encouraging women through education is recommended to increase health facility delivery service utilization.

## Acknowledgments

We greatly acknowledge MEASURE DHS for granting access to the East African DHS data sets.

## Author Contributions

**Conceptualization:** Getayeneh Antehunegn Tesema, Zemenu Tadesse Tessema.

**Data curation:** Getayeneh Antehunegn Tesema, Zemenu Tadesse Tessema.

**Formal analysis:** Getayeneh Antehunegn Tesema, Zemenu Tadesse Tessema.

**Investigation:** Getayeneh Antehunegn Tesema, Zemenu Tadesse Tessema.

**Methodology:** Getayeneh Antehunegn Tesema, Zemenu Tadesse Tessema.

**Software:** Getayeneh Antehunegn Tesema, Zemenu Tadesse Tessema.

**Validation:** Getayeneh Antehunegn Tesema, Zemenu Tadesse Tessema.

**Visualization:** Getayeneh Antehunegn Tesema, Zemenu Tadesse Tessema.

**Writing – original draft:** Getayeneh Antehunegn Tesema.

**Writing – review & editing:** Getayeneh Antehunegn Tesema, Zemenu Tadesse Tessema.

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
