## [Decision Letter · Decision Letter 0]

28 Sep 2020

PONE-D-20-19261

Pooled prevalence and determinants of health facility delivery in East Africa: A pooled analysis of Demographic and Health Surveys

PLOS ONE

Dear Dr. Tesema,

Thank you for submitting your manuscript to PLOS ONE. After careful consideration, we feel that it has merit but does not fully meet PLOS ONE’s publication criteria as it currently stands. Therefore, we invite you to submit a revised version of the manuscript that addresses the points raised during the review process.

The reviewers have provided extensive feedback on improving the rational, methods, and discussion. Please carefully respond to each comment. Once you have completed the revisions, please have your manuscript professionally edited.

We look forward to receiving your revised manuscript.

Kind regards,

Nancy Beam, PhD

Senior Editor

PLOS ONE

Journal Requirements:

2.In your Data Availability statement, you have not specified where the minimal data set underlying the results described in your manuscript can be found. PLOS defines a study's minimal data set as the underlying data used to reach the conclusions drawn in the manuscript and any additional data required to replicate the reported study findings in their entirety. All PLOS journals require that the minimal data set be made fully available. For more information about our data policy, please see http://journals.plos.org/plosone/s/data-availability.

Reviewers' comments:

Reviewer's Responses to Questions

**Comments to the Author**

1. Is the manuscript technically sound, and do the data support the conclusions?

Reviewer #1: Partly

Reviewer #2: Yes

2. Has the statistical analysis been performed appropriately and rigorously? 

Reviewer #1: No

Reviewer #2: Yes

3. Have the authors made all data underlying the findings in their manuscript fully available?

Reviewer #1: No

Reviewer #2: Yes

4. Is the manuscript presented in an intelligible fashion and written in standard English?

Reviewer #1: No

Reviewer #2: Yes

5. Review Comments to the Author

Reviewer #1: I would like to thank the authors for their submission on this important topic. In summary, this manuscript requires assistance from a native English speaker to improve the grammar and sentence structure, as well as the organization of the material. I found these issues to be rather distracting while reading the manuscript. Secondly, I have made suggestions for improving different sections of the manuscript below. I have also included some suggested rewrites for some statements in the manuscript.

Introduction

The authors did not make a clear case for the importance of a pooled study. Decision-making is typical at national and sub-national levels in individual countries so it is not clear what a pooled prevalence study adds. How would a study like this influence policy within each country? The question of “why is this important” is not addressed.

Methods

Data Source: Please provide a citation for your data sources, specifically, the DHS program

Please provide a description of your dataset. How many women responded to each question by country? How many responded “Yes” or “No”? How much missing data did you have? What was the DHS year for each country’s dataset? How did you decide on the independent variables?

Please conduct sensitivity analysis to investigate the impact of having more participants from Kenya in this. How does this influence the study findings?

Discussion

This section needs to be better organized. The discussion should be used to summarize the study findings, discuss its implications and make recommendations. What does the study findings imply for progress towards the 2030 SDGs? What can be done to improve health facility delivery in East Africa based on the study findings? While it is important to highlight how the study findings is related to existing literature, it is important that actionable insights are drawn from the study

Lines 212 – 214: It seems the aims of the study as stated here differs from what was declared in the introduction

Suggestions for improving grammar

Lines 54 – 56: Maternal mortality has decreased significantly in developed countries (3) but SSA continued to account for 66% of maternal deaths worldwide

Suggestion: “continued” should be continues

Lines 58 – 59: As key strategies for reducing maternal and infant mortality, the World Health Organization (WHO) recommended a health facility delivery

Suggested rewrite: The World Health Organization recommends health facility delivery as a key strategy for reducing maternal and infant mortality

Lines 63 – 64: The proportion of institutional delivery serves as a measure of progress towards maternal and infant mortality reduction

Suggested rewrite: The proportion of institutional delivery serves as a measure of progress towards reductions in maternal and infant mortality.

Lines 64 – 67: The East African countries that include Burundi, Ethiopia, Comoros, Uganda, Rwanda, Tanzania, Mozambique, Madagascar, Zimbabwe, Kenya, Zambia, and Malawi are among the world's poor countries with accessibility and affordability of maternal health care services

Suggested rewrite: Globally, East African countries rank the lowest for affordability and accessibility of maternal health services, and have the greatest share of maternal deaths.

Line 175: “was” should be “were”

Reviewer #2: Reviewer feedback

The authors make an important contribution to our understanding of the factors associated with health facility delivery. The main advantages of this paper are 1) it uses pooled DHS data from multiple countries, which is a new approach, and 2) the methodological approach is robust and appropriate. The paper is very well organized and written is terms of structure, flow of ideas and coherence. However, there are numerous language errors and awkward sentences so the paper could benefit from professional editing.

Detailed comments

1. I think pooling DHS data is an important contribution of this paper. However, the authors do not say this approach is useful or worth pursing in the paper. The authors should explicitly and clearly explain this.

2. It is not clear what are the years for these datasets? Are you all for the same year or difference years? And what did the authors do to make sure that the time dimension is not influencing the results?

3. There are issues with merging DHS datasets from different countries related to the use of DHS instrument itself in each country and data consistency issues that must be at least covered in the limitations section

4. Some of the statements are missing a source/reference. For example, Page 10, line 215, the reference for the mentioned study is not listed.

5. Page 11, line 246, “The potential reason could be that educated husbands can include women making decisions about the use of maternal health services in health care.” The way this is written implies that decisions about women’s health are naturally the responsibility of men but educated men include women in the decision. I do not think this is what the authors intended to say so they should write this point more carefully.

6. I found that the weakest part of the paper are the discussion and more so the conclusions. The discussion should go beyond simply saying when the results agree and disagree with other studies.

7. The study finds that “country, urban residence, maternal education, husband education, multiple gestations, wanted pregnancy, ANC visit during pregnancy, middle and rich wealth status, having an occupation, and married marital status was positively associated with health facility delivery.” These results are hardly surprising; this is what we know already. The study only confirms what we already know. The authors should highlight how they believe this study adds value and contributes to better understanding.

8. The conclusions are quite vague and not helpful. The authors should discuss policy implications of their work. “Therefore, the governmental and non-governmental organizations should scale up their programs to encourage women education and ANC service utilization for pregnant women.” This statement is completely generic and unhelpful. Why should non-governmental organizations listen? An adequate well-written policy implication section should be included in the conclusions section.

9. Language

• Page 2, line 17 – “But still….” Is an informal way to express this idea

• Page 3, line 44- “ the government and non-governmental…. what? – awkward sentence”

• Page 10, line 218-219, “This could be due to the Ethiopian health system suffers from a lack of adequate medical care and human resources to meet more” There are grammatical errors in this sentence, and it seems like shortage or inadequate supply would be more accurate than “lack” which means there are none.

• Page 12, line 252, “women who were not in union” this is not a clear way to sayunmarried.

• Page 12, line 254 “This might be due to married women had spousal support in making health care decisions towards maternal health service utilization as well as economic and social support” grammatical errors.

• Page 14, line 297, it is better not to say “can’t” and “didn’t” – can not and did not

• Page 14, line 303, “showed that there is the health facility delivery utilization by the reproductive age women has been significantly varied across countries in East Africa.” Sente

6. PLOS authors have the option to publish the peer review history of their article (what does this mean?). If published, this will include your full peer review and any attached files.

Reviewer #1: **Yes: **Ifeoma D Ozodiegwu

Reviewer #2: No

---

## [Author Response · Author response to Decision Letter 0]

26 Oct 2020

Point by point response 

Manuscript Title: Pooled prevalence and determinants of health facility delivery in East Africa: A pooled analysis of Demographic and Health Surveys/ Pooled prevalence and associated factors of health facility delivery in East Africa: A pooled analysis of Demographic and Health Surveys

Manuscript ID: PONE-D-20-19261

Dear editor/reviewers:

Thank you for giving us the chance to revise the manuscript. The comments are too imperative, which are important for improving the quality of our paper. We have addressed all of the concerns raised and these modifications are also incorporated in the revised manuscript.

Response to reviewers’ comment 

1. I would like to thank the authors for their submission on this important topic. In summary, this manuscript requires assistance from a native English speaker to improve the grammar and sentence structure, as well as the organization of the material. I found these issues to be rather distracting while reading the manuscript. Secondly, I have made suggestions for improving different sections of the manuscript below. I have also included some suggested rewrites for some statements in the manuscript.

Authors’ response: Thank you for the comments. We extensively modified the sentence structure and any typographical errors with the help of experts. Besides, we consider your suggestions as well as your constructive comments and we modified the manuscript. (See the revised manuscript)

2. Introduction, the authors did not make a clear case for the importance of a pooled study. Decision-making is typical at national and sub-national levels in individual countries so it is not clear what a pooled prevalence study adds. How would a study like this influence policy within each country? The question of “why is this important” is not addressed.

Authors’ response: Thank you reviewer for the comments. As you stated very well decision-making is typical at the national and sub-national level in each country, considering this issue we have reported the prevalence of health facility delivery for each country as well we have reported the prevalence of health facility delivery at the East Africa level to know what health facility level looks like as compared to other regions. Just the main thing we use pooled analysis is not only to know the pooled prevalence but also to increase the statistical power of the study to detect the true effect of independent variables. As you know when we pool the DHSs of the 12 East African countries, the sample size increases and could result in the corresponding improvement in the power of the study and we would get valid evidence. Besides, the DHS of the 12 East African countries uses similar study design and sampling procedures, and therefore the estimate we got in this study is reliable to use for policymakers. (See the background and Discussion section, line 98-101, and line 246-250, page 5 and 11)

3. Methods, data Source: Please provide a citation for your data sources, specifically, the DHS program

Authors’ response: Thank you reviewer for the comments. We cited the DHS program address. (See the revised manuscript, line 107, and page 5)

4. Please provide a description of your dataset. How many women responded to each question by country? How many responded “Yes” or “No”? How much missing data did you have? What was the DHS year for each country’s dataset? How did you decide on the independent variables?

Authors’ response: Thank you reviewer for the comments. There was no missing as we kept our study participants as women who gave birth within five years preceding the survey and present the study sample in each country through the table. We retrieved the independent variables from the DHSs based on literature. Besides, we select variables for the final model using the LASSO and Elastic Net method in addition to the bivariable analysis. (See Table 1)

5. Please conduct a sensitivity analysis to investigate the impact of having more participants from Kenya on this. How does this influence the study findings?

Authors’ response: Thank you reviewer for the comments. We reported the prevalence using forest plots but the study was not a meta-analysis. This study was a pooled data analysis, just to know the prevalence of health facility delivery and associated factors in East Africa and in each country based on the DHSs data that have the same design as well as sampling procedure. So, our aim is not to get the pooled estimate like in metanalysis and the I-squared reported in this is not to show the heterogeneity. Here in this study, the unit of analysis was individuals but not studies like a meta-analysis. Overall, our study aimed to use to pooled DHS data to increase the statistical power of the study to detect the true effect of the variables. Besides, we aimed to assess whether health facility delivery utilization has been varied across countries in East Africa, and if we remove Kenya, we can not answer it. Furthermore, we have checked whether the estimates varied with the presence and absence of Kenya, and the result showed there was a difference in the prevalence of health facility delivery but not in the OR.

6. Discussion, this section needs to be better organized. The discussion should be used to summarize the study findings, discuss its implications and make recommendations. What does the study findings imply for progress towards the 2030 SDGs? What can be done to improve health facility delivery in East Africa based on the study findings? While it is important to highlight how the study findings is related to existing literature, it is important that actionable insights are drawn from the study, Lines 212 – 214: It seems the aims of the study as stated here differs from what was declared in the introduction.

Authors’ response: Thank you reviewer for the comments. We summarize the study findings and their implications in the Discussion section of the manuscript. Besides, we compare our findings with previous studies and we wrote out the possible explanations. (See the revised manuscript)

7. Suggestions for improving grammar

- Lines 54 – 56: Maternal mortality has decreased significantly in developed countries (3) but SSA continued to account for 66% of maternal deaths worldwide

Suggestion: “continued” should be continues

Authors’ response: Thank you reviewer we modified it. (See the Background section, line 67, page4)

- Lines 58 – 59: As key strategies for reducing maternal and infant mortality, the World Health Organization (WHO) recommended a health facility delivery

Suggested rewrite: The World Health Organization recommends health facility delivery as a key strategy for reducing maternal and infant mortality

Authors’ response: Thank you reviewer for the comments. We rewrite it. (See the background section, line 70-71, page 4)

- Lines 63 – 64: The proportion of institutional delivery serves as a measure of progress towards maternal and infant mortality reduction

Suggested rewrite: The proportion of institutional delivery serves as a measure of progress towards reductions in maternal and infant mortality.

Authors’ response: Thank you reviewer for the comments. We rewrite it. (See the Background section, line 71-72, page 4)

- Lines 64 – 67: The East African countries that include Burundi, Ethiopia, Comoros, Uganda, Rwanda, Tanzania, Mozambique, Madagascar, Zimbabwe, Kenya, Zambia, and Malawi are among the world's poor countries with accessibility and affordability of maternal health care services

Suggested rewrite: Globally, East African countries rank the lowest for affordability and accessibility of maternal health services, and have the greatest share of maternal deaths.

Authors’ response: Thank you reviewer for your suggestion. We rewrite it. (See Background section, line 77-79, page 4)

- Line 175: “was” should be “were”

Authors’ response: Thank you reviewer for the suggestion. We rewrite it. (see the discussion section, line 263, page 12)

Response to Reveiwer#2

The authors make an important contribution to our understanding of the factors associated with health facility delivery. The main advantages of this paper are 1) it uses pooled DHS data from multiple countries, which is a new approach, and 2) the methodological approach is robust and appropriate. The paper is very well organized and written is terms of structure, flow of ideas and coherence. However, there are numerous language errors and awkward sentences so the paper could benefit from professional editing.

Authors’ response: Thank you, reviewer. We extensively modified the body of the manuscript for any language error with the help of language experts at the University of Gondar. (See the revised manuscript)

1. I think pooling DHS data is an important contribution of this paper. However, the authors do not say this approach is useful or worth pursing in the paper. The authors should explicitly and clearly explain this.

Authors’ response: Thank you reviewer for your comments to strengthen our paper. We used pooled data analysis for this study. As you stated, pooling DHS data is very important to increase the statistical power of the study to detect the true effect of the independent variables in addition to providing the prevalence of health facility delivery at the East Africa level as well as in each country. This is important for WHO as well for international programs to design targeted interventions. Besides, we used the advanced statistical analysis technique that was multilevel analysis, this could be helpful to provide valid evidence for policymakers and program planners. We stated in the background and discussion section in a short and precise manner. (See the background and Discussion section, line 98-101, and line 246-250, page 5 and 11)

2. It is not clear what are the years for these datasets? Are you all for the same year or difference years? And what did the authors do to make sure that the time dimension is not influencing the results?

Authors’ response: Thank you reviewer for the comments. The years for these datasets were from 2008/09 to 2018, and to detect whether the time dimension was influencing the study results by categorizing years into two as 0 “If the study period ≤2015” and 1 “if the study period was > 2015” the categorization was based on the millennium development goal as during MDG and during SDG. But unfortunately, it was not eligible for the final model as it has a p-value greater than 0.2 in the bi-variable analysis. Besides, we have done a chi-square test whether the year of the survey was significantly associated with health facility delivery but it was not significant. (See Table 1)

3. There are issues with merging DHS datasets from different countries related to the use of DHS instrument itself in each country and data consistency issues that must be at least covered in the limitations section.

Authors’ response: Thank you reviewer for the comments. The DHS uses similar measurements for the variables in each country and the data are consistent. We have checked the consistency of measurement of variables before we do the analysis and the variables were consistently measured. Besides, the variables we have used in this study may not need that much-sophisticated measurement as it was self-report from the respondents through interviews. As a limitation, we have stated the issues related to the study design used in DHS and the variables missed in DHS but that was important for this study. 

4. Some of the statements are missing a source/reference. For example, on Page 10, line 215, the reference for the mentioned study is not listed.

Authors response: Thank you reviewer for the comments. We provide an appropriate citation for the statements we have used in the manuscript. (See the revised manuscript)

5. Page 11, line 246, “The potential reason could be that educated husbands can include women making decisions about the use of maternal health services in health care.” The way this is written implies that decisions about women’s health are naturally the responsibility of men but educated men include women in the decision. I do not think this is what the authors intended to say so they should write this point more carefully.

Authors’ response: Thank you reviewer for the comments. We rewrite it. (see the Discussion section, line 283-289, page 13)

6. I found that the weakest part of the paper is the discussion and more so the conclusions. The discussion should go beyond simply saying when the results agree and disagree with other studies

Authors’ response: Thank you reviewer for the comments. We rewrite it extensively. (See the revised manuscript)

7. The study finds that "country, urban residence, maternal education, husband education, multiple gestations, wanted pregnancy, ANC visit during pregnancy, middle and rich wealth status, having an occupation, and married marital status was positively associated with health facility delivery." These results are hardly surprising; this is what we know already. The study only confirms what we already know. The authors should highlight how they believe this study adds value and contributes to a better understanding.

Authors’ response: Thank you reviewer for the comments. We had stated in the discussion and conclusion section of the study. This study adds both statistical and public health value. Regarding statistical value, this study was done by pooling DHSs data conducted in 12 East African countries and this increases the statistical power of the study to detect the true effect size. Besides, we applied the mixed-effect analysis as since DHS data has hierarchical nature, therefore, the findings are reliable as it was analyzed using the advanced model. Regarding the public health aspect, this study was conducted at the East Africa level and we get the pooled estimate and the pooled estimate of health facility delivery in East Africa and the prevalence of health facility delivery for each country as well as factors associated with it. These findings are reliable as it was based on weighted large data using the advanced model and this could help to design evidence-based international and national public health programs.

8. The conclusions are quite vague and not helpful. The authors should discuss policy implications of their work. “Therefore, the governmental and non-governmental organizations should scale up their programs to encourage women education and ANC service utilization for pregnant women.” This statement is completely generic and unhelpful. Why should non-governmental organizations listen? An adequate well-written policy implication section should be included in the conclusions section.

Authors’ response: Thank you reviewer for the comments. We rewrite it. (See the conclusion section, line 360-374, page 16)

9. Language

• Page 2, line 17 – “But still….” Is an informal way to express this idea

• Page 3, line 44- “ the government and non-governmental…. what? – awkward sentence”

• Page 10, line 218-219, “This could be due to the Ethiopian health system suffers from a lack of adequate medical care and human resources to meet more” There are grammatical errors in this sentence, and it seems like shortage or inadequate supply would be more accurate than “lack” which means there are none.

• Page 12, line 252, “women who were not in union” this is not a clear way to sayunmarried.

• Page 12, line 254 “This might be due to married women had spousal support in making health care decisions towards maternal health service utilization as well as economic and social support” grammatical errors.

• Page 14, line 297, it is better not to say “can’t” and “didn’t” – can not and did not

• Page 14, line 303, “showed that there is the health facility delivery utilization by the reproductive age women has been significantly varied across countries in East Africa.” Sente

Authors’ response: Thank you reviewer for your constructive comments. We extensively modified the document by taking into your comments. (See the revised manuscript)

---

## [Editor Report · Decision Letter 1]

12 Jan 2021

PONE-D-20-19261R1

Pooled prevalence and associated factors of health facility delivery in East Africa: A pooled analysis of Demographic and Health Surveys

PLOS ONE

Dear Dr. Tesema,

Thank you for submitting your manuscript to PLOS ONE. After careful consideration, we feel that it has merit but does not fully meet PLOS ONE’s publication criteria as it currently stands. Therefore, we invite you to submit a revised version of the manuscript that addresses the points raised during the review process.

We look forward to receiving your revised manuscript.

Kind regards,

Marwa Farag

Academic Editor

PLOS ONE

The following concerns were not adequately addressed by the author:

The authors addressed some of the points raised by the reviewers adequately. However, some key points and concerns were not addressed in the revised draft.

1. What did the authors do to make sure that the time dimension is not influencing the results?

The authors stated that the DHS dataset years spanned from 2008 to 2018. This is a long time period and they do not provide convincing evidence that year of study did not matter. The approach they adopted of picking a random year 2015 and testing it whether there is a significant difference before or after is not sufficient. Have you considered other approaches, including year fixed effects (dummies) or any other appropriate approach?

2. What is the justification for adopting this design of a pooled analysis?

The statistical justification for adopting the pooled design is not sufficient. The DHS datasets are already large enough. The justification should include meaningful reasons, such as understanding factors common or that apply across the region. The authors should be able to answer why it is a good idea to do this work and the answer cannot just be to increase statistical power.

3. Language use in the manuscript is still inadequate. Here are examples from the abstract:

Page 2, line 18 – “but still home delivery is common in” – This is an informal way to say this.

Page 3, line 53 - “Was far below to achieve” – Sentence structure issue

The entire conclusions section is not well written. There are many awkward sentences such as: “Moreover, special attention should be needed for poor”

This manuscript needs to edited by a professional editor. It does not matter if the editor is a native english speaker or not. It needs professional editing by a professional.

---

## [Author Response · Author response to Decision Letter 1]

26 Feb 2021

Point by point response for editors/reviewers comments 

PLOS ONE Journal 

Manuscript title: Pooled prevalence and associated factors of health facility delivery in East Africa: A pooled analysis of Demographic and Health Surveys/ Pooled prevalence and associated factors of health facility delivery in East Africa: Mixed-effect logistic regression analysis

Manuscript ID: PONE-D-20-19261R1

Dear editor/reviewer. 

Dear all,

We would like to thank you for these constructive, building, and improvable comments on this manuscript that would improve the substance and content of the manuscript. We considered each comment and clarification questions of editors and reviewers on the manuscript thoroughly. Our point-by-point responses for each comment and question are described in detail on the following pages. Further, the details of changes were shown by track changes in the supplementary document attached.

Response to Editors’ comment 

1. The following concerns were not adequately addressed by the author:

The authors addressed some of the points raised by the reviewers adequately. However, some key points and concerns were not addressed in the revised draft.

Authors’ response: Thank you for the comments, we have addressed all the comments. (See the revised manuscript)

2. What did the authors do to make sure that the time dimension is not influencing the results?

The authors stated that the DHS dataset years spanned from 2008 to 2018. This is a long period and they do not provide convincing evidence that year of study did not matter. The approach they adopted of picking a random year 2015 and testing it whether there is a significant difference before or after is not sufficient. Have you considered other approaches, including year fixed effects (dummies) or any other appropriate approach?

Authors’ response: Thank you for the comments. We run the model considering the year of the survey as a fixed effect (dummies) considering 2008 as a reference but it was not eligible for the final analysis as it has a p-value < 0.2.

We generate a variable year of the survey as, 2008 (Madagascar), 2011 (Mozambique), 2012 (Malawi), 2014 (Rwanda and Kenya), 2015 (Malawi, Tanzania, and Zimbabwe), 2016 (Burundi, Uganda, and Ethiopia), and 2018 (Zambia). Descriptively the prevalence over time was ranging from 44.5% to 72.5% showed a change over time but the regression was not significant when we considered it as a dummy variable and adjusted with the presence of other predictors. If there is a need to report in the manuscript, we are ready to incorporate it. 

3. What is the justification for adopting this design of a pooled analysis?

The statistical justification for adopting the pooled design is not sufficient. The DHS datasets are already large enough. The justification should include meaningful reasons, such as understanding factors common or that apply across the region. The authors should be able to answer why it is a good idea to do this work and the answer cannot just be to increase statistical power.

Authors’ response: Thank you for the comments. We provide additional justification considering the direction you provide us. Previously we justified it with respect to the statistical significance of pooled analysis but now based on your critical suggestion we justify the clinical implications of a pooled analysis. (See the revised manuscript)

4. Language use in the manuscript is still inadequate. Here are examples from the abstract:

Page 2, line 18 – “but still home delivery is common in” – This is an informal way to say this.

Page 3, line 53 - “Was far below to achieve” – Sentence structure issue

The entire conclusions section is not well written. There are many awkward sentences such as: “Moreover, special attention should be needed for poor”

This manuscript needs to edited by a professional editor. It does not matter if the editor is a native english speaker or not. It needs professional editing by a professional.

Authors’ response: Thank you for the comments. We extensively modified the entire document with the help of language experts at the university. (See the revised manuscript)

---

## [Editor Report · Decision Letter 2]

9 Mar 2021

PONE-D-20-19261R2

Pooled prevalence and associated factors of health facility delivery in East Africa: Mixed-effect logistic regression analysis

PLOS ONE

Dear Dr. Tesema ,

Thank you for submitting your manuscript to PLOS ONE. After careful consideration, we feel that it has merit but does not fully meet PLOS ONE’s publication criteria as it currently stands. Therefore, we invite you to submit a revised version of the manuscript that addresses the points raised during the review process.

We look forward to receiving your revised manuscript.

Kind regards,

Marwa Farag

Academic Editor

PLOS ONE

Journal Requirements:

Additional Editor Comments (if provided):

The authors have addressed the major comments in the paper. However, there are a few issues that still need to be addressed:

1- There are still language errors in the document. Even the first few lines of the abstract have several errors and I am not sure what pocket studies mean?:

"Despite health facility delivery is identified as a key strategy for reducing

maternal and neonatal mortality, it less utilized in many African countries. There are

pocket studies on the prevalence and associated factors of health facility delivery in

different East African countries but the prevalence and significant factors were varied

from study to study."

"So, pooled analysis using the nationally representative DHS data of East African

114 countries is vital for understanding associated factors common across countries." Is an informal way to express this idea.

This paper MUST be reviewed by a professional editor.

2- The discussion about how the time dimension was handled also need to be included in the manuscript.

---

## [Author Response · Author response to Decision Letter 2]

12 Mar 2021

Point by point response for editors/reviewers comments 

PLOS ONE Journal 

Manuscript title: Pooled prevalence and associated factors of health facility delivery in East Africa: Mixed-effect logistic regression analysis

Manuscript ID: PONE-D-20-19261R2

Dear editor. 

Dear all,

We would like to thank you for these constructive, building, and improvable comments on this manuscript that would improve the substance and content of the manuscript. We considered each comment and clarification questions of editors and reviewers on the manuscript thoroughly. Our point-by-point responses for each comment and question are described in detail on the following pages. Further, the details of changes were shown by track changes in the supplementary document attached.

Response to Editors’ comment 

1. - There are still language errors in the document. Even the first few lines of the abstract have several errors and I am not sure what pocket studies mean?:

"Despite health facility delivery is identified as a key strategy for reducing

maternal and neonatal mortality, it less utilized in many African countries. There are

pocket studies on the prevalence and associated factors of health facility delivery in

different East African countries but the prevalence and significant factors were varied

from study to study." "So, pooled analysis using the nationally representative DHS data of East African 114 countries is vital for understanding associated factors common across countries." Is an informal way to express this idea.

Authors’ response: Thank you for the comments. We extensively modified the sentence structure and for any typographical error with the help of language experts at the university. (See the revised manuscript)

2. The discussion about how the time dimension was handled also need to be included in the manuscript.

Authors’ response: Thank you for the comments. We included in the method section. (See the revised manuscript)

---

## [Editor Report · Decision Letter 3]

16 Mar 2021

PONE-D-20-19261R3

Pooled prevalence and associated factors of health facility delivery in East Africa: Mixed-effect logistic regression analysis

PLOS ONE

Dear Dr. Tesema,

Thank you for submitting your manuscript to PLOS ONE. After careful consideration, we feel that it has merit but does not fully meet PLOS ONE’s publication criteria as it currently stands. Therefore, we invite you to submit a revised version of the manuscript that addresses the points raised during the review process.

We look forward to receiving your revised manuscript.

Kind regards,

Marwa Farag

Academic Editor

PLOS ONE

Journal Requirements:

Additional Editor Comments (if provided):

There are still language errors even in the abstract:

Examples:

1- "This study showed that health facility delivery in East African countries is low." It should say the percentage or rate of health facility delivery)

2- "These findings suggested that maternal and child health programs should enhance health facility delivery in rural residents and poor households by enhancing maternal education, and ANC service utilization." This sentence is poorly written and hard to understand.

Please have the manuscript professionally edited. There are still errors even in the abstract.

---

## [Author Response · Author response to Decision Letter 3]

20 Mar 2021

Point by point response for editors/reviewers comments 

PLOS ONE Journal 

Manuscript title: Pooled prevalence and associated factors of health facility delivery in East Africa: Mixed-effect logistic regression analysis

Manuscript ID: PONE-D-20-19261R3

Dear editor. 

Dear all,

We would like to thank you for these constructive, building, and improvable comments on this manuscript that would improve the substance and content of the manuscript. We considered each comment and clarification questions of editors on the manuscript thoroughly. Our point-by-point responses for each comment and question are described in detail on the following pages. Further, the details of changes were shown by track changes in the supplementary document attached.

Response to Editors’ comment 

1. There are still language errors even in the abstract:

Examples:

1- "This study showed that health facility delivery in East African countries is low." It should say the percentage or rate of health facility delivery)

2- "These findings suggested that maternal and child health programs should enhance health facility delivery in rural residents and poor households by enhancing maternal education, and ANC service utilization." This sentence is poorly written and hard to understand.

Please have the manuscript professionally edited. There are still errors even in the abstract.

Authors’ response: Thank you Editor for the comments. We extensively edited the whole manuscript for any typographical and grammatical errors. Besides, it is reviewed by language experts at the university. (See the revised manuscript)

---

## [Editor Report · Decision Letter 4]

24 Mar 2021

PONE-D-20-19261R4

Pooled prevalence and associated factors of health facility delivery in East Africa: Mixed-effect logistic regression analysis

PLOS ONE

Dear Dr. Tesema,

Thank you for submitting your manuscript to PLOS ONE. After careful consideration, we feel that it has merit but does not fully meet PLOS ONE’s publication criteria as it currently stands. Therefore, we invite you to submit a revised version of the manuscript that addresses the points raised during the review process.

We look forward to receiving your revised manuscript.

Kind regards,

Marwa Farag

Academic Editor

PLOS ONE

Journal Requirements:

Additional Editor Comments (if provided):

The language of the manuscript is still inadequate. The manuscript cannot be accepted in its current form.

There are still language errors - even in the abstract.

This is the conclusions section in the abstract of the paper:

Conclusion: This study showed that the proportion of health facility delivery in East African

countries is low.

*Clear 

'Country, residence, maternal education status, husband education status,

marital status, occupation status, wealth status, ANC visit, health care access, parity, type of

gestation and wanted pregnancy were significantly associated factors of health facility delivery.'

*This sentence is typically reported as part of the results and not conclusions section

Enhancing maternal education and ANC visit would increase delivery at a health facility. We

recommended maternal health programs targeting improving health facility delivery should emphasize for rural residents.

* You cannot say enhance ANC visit? what does this mean? Also would increase delivery at a health facility implies that you established causality, which is not the case so you should say likely to increase or expected to increase instead. 

* the sentence "We recommended maternal health programs targeting improving health facility delivery should emphasize for rural residents." is grammatically incorrect and poorly structured

---

## [Author Response · Author response to Decision Letter 4]

29 Mar 2021

PLOS ONE 

Point by point response for editors comments 

Manuscript title: Pooled prevalence and associated factors of health facility delivery in East Africa: Mixed-effect logistic regression analysis

Manuscript ID: PONE-D-20-19261R4

Dear editor/reviewer. 

Dear all,

We would like to thank you for these constructive, building, and improvable comments on this manuscript that would improve the substance and content of the manuscript. We considered each comment on the manuscript thoroughly. Our point-by-point responses for each comment and question are described in detail on the following pages.

Response to Editors comment

1. Journal Requirements:

Authors’ response: Thank you for the concerns. We assessed all the references cited in the manuscript and there is no retracted refrences.

2. The language of the manuscript is still inadequate. The manuscript cannot be accepted in its current form.

There are still language errors - even in the abstract.

This is the conclusions section in the abstract of the paper:

Conclusion: This study showed that the proportion of health facility delivery in East African

countries is low.

*Clear 

'Country, residence, maternal education status, husband education status,

marital status, occupation status, wealth status, ANC visit, health care access, parity, type of

gestation and wanted pregnancy were significantly associated factors of health facility delivery.'

*This sentence is typically reported as part of the results and not conclusions section

Enhancing maternal education and ANC visit would increase delivery at a health facility. We

recommended maternal health programs targeting improving health facility delivery should emphasize for rural residents.

* You cannot say enhance ANC visit? what does this mean? Also would increase delivery at a health facility implies that you established causality, which is not the case so you should say likely to increase or expected to increase instead. 

* the sentence "We recommended maternal health programs targeting improving health facility delivery should emphasize for rural residents." is grammatically incorrect and poorly structured

Authors’ response: Thank you for the commnets. We extensively edited and corrected the sentence structure with the help of language experts at the university. (See the revised manuscript)

---

## [Editor Report · Decision Letter 5]

7 Apr 2021

Pooled prevalence and associated factors of health facility delivery in East Africa: Mixed-effect logistic regression analysis

PONE-D-20-19261R5

Dear Dr. Tesema,

We’re pleased to inform you that your manuscript has been judged scientifically suitable for publication and will be formally accepted for publication once it meets all outstanding technical requirements.

Kind regards,

Marwa Farag

Guest Editor

PLOS ONE
---

## [Editor Report · Acceptance letter]

13 Apr 2021

PONE-D-20-19261R5 

Pooled prevalence and associated factors of health facility delivery in East Africa: Mixed-effect logistic regression analysis 

Dear Dr. Tesema:

I'm pleased to inform you that your manuscript has been deemed suitable for publication in PLOS ONE. Congratulations! Your manuscript is now with our production department. 

Kind regards, 

on behalf of

Dr. Marwa Farag 

Guest Editor

PLOS ONE